# Effectiveness of Agreement Criteria and Flows of Collaborative Care in Primary Mental Health Care in Brazil

**DOI:** 10.3390/ijerph192215148

**Published:** 2022-11-17

**Authors:** Valdecir Carneiro da Silva, Ardigleusa Alves Coêlho, Ana Angélica Rêgo de Queiroz, Gabriela Maria Cavalcanti Costa, João Carlos Alchieri, Ricardo Alexandre Arcêncio, Severina Alice da Costa Uchôa

**Affiliations:** 1Department of Nursing, State University of Paraiba, Rua José do Ó, 596, Campina Grande 58401-411, Brazil; 2Rio Grande do Norte Research Support Foundation, Natal 59078-970, Brazil; 3Department of Psychology, Federal University of Rio Grande do Norte, Natal 59078-970, Brazil; 4College of Nursing, University of São Paulo, Ribeirão Preto 05508-060, Brazil; 5Department of Collective Health, Federal University of Rio Grande do Norte, Natal 59078-970, Brazil

**Keywords:** mental health, primary health care, collaborative care, health assessment, pharmacovigilance

## Abstract

The supply of mental health processes in primary care has gaps. This study aims to analyze the association of agreement criteria and flows between primary care teams and the Family Health Support Center (NASF) for mental health collaborative care, considering the difference between capital and non-capital cities in Brazil. This cross-sectional study was conducted based on secondary data from the Primary Care Access and Quality Improvement Program. Agreement criteria and flows were obtained from 3883 NASF teams of the matrix support or collaborative care. The Chi-square test and multiple Poisson regression were used; *p* < 0.05 was considered statistically significant. Prevalence ratios of negative associations demonstrated protective factors for support actions: follow-up at Psychosocial Care Center, management of psychopharmacotherapy, offer of other therapeutic actions, care process for users of psychoactive substances, and offer of activities to prevent the use of psychoactive substances. Collaborative care in primary care was effective, and capital cities were a protective factor compared with non-capital cities.

## 1. Introduction

Specific approaches of Health Care Networks, such as the psychosocial care network (PCN), are a preferential gateway to the primary health care (PHC) of the Brazilian Unified Health System (SUS). These approaches coordinate care and integrate mental health between PHC services and specialized care [1]. Despite this, providing quality care to people with mental and behavioral disorders is challenging in global health systems due to high mortality rates and reduced life expectancy [2].

The World Health Organization (WHO) World Report on Mental Health highlighted that one in eight people have some type of mental disorder and its prevalence varies according to sex and age. Mental disorders are the main causes of years lived with disability (YLD) and represent one in six cases of YLD worldwide [3]. In the current context of the COVID-19 pandemic, the global crisis has worsened and expanded, weakening services, and increasing the therapeutic gap for mental health [3].

The reduced integration of Health Care Networks and lack of qualified professionals to offer mental health care processes in the PHC leads to gaps in diagnostic accuracy and treatment, overload of specialized services, and difficulty accessing and responding to demands. Strategies used to overcome these difficulties are the implementation of devices to integrate Health Care Networks and investment in partnerships between PHC professionals and specialists via matrix support or collaborative care, specifically the PCN [2].

Mental health collaborative care emerged in the United States of America (New York) in the 1970s through the Gouverneur Health Services Program, which joined the Health Maintenance Organization Program of the Community Health Care Plan (New Haven, Connecticut) [4,5]. Collaborative care includes individual, interprofessional, and organizational interventions to improve coordination and quality of care [6]. Australia, Canada, Cuba, France, Portugal, Norway, the United Kingdom, the Netherlands, and other Latin American countries have also used collaborative care or matrix support to qualify PHC and expand the offer of primary mental health care (organization of services and clinical practices) [7].

The matrix support is a collaborative care model created in the Brazilian mental health context [8]. This model is offered by the Family Health Support Center (NASF) and primary care teams (PCT) through the Family Health Strategy (FHS) and is necessary for the integration of mental health into PHC. Internationally, this type of care is called shared care or consultation-liaison [9].

The World Health Organization (WHO) showed that only 25% of member countries met the criteria for integrating mental health into PHC. Despite the expansion of mental health promotion and prevention programs, the goals of the Comprehensive Mental Health Action Plan were extended to 2030 [10]. Thus, WHO guidelines for mental health policies, planning, and programs recommend actions to integrate PHC services into the person-centered care model with respect and protection of human rights [11,12].

The Collaborative Care Model (CoCM) is a well-known strategy, based on evidence, grounded on a Chronical Care Model (CCM), and was originally made to describe complex evidence related to the care level to (i) accomplish a multiprofessional approach; (ii) improve interprofessional communication; (iii) enhance programmed health follow-ups; and (iv) make the structured management plan [13,14,15]. Regarding health care procedures, those apply the care management approach by systematic monitoring and links between primary care and mental health professionals, offering patients: (1) mental health exams; (2) interventions; (3) care facilitations; and (4) follow-ups [16].

In the current epidemiological transition, impacts prevail. Before and after the COVID-19 pandemic, mental and behavioral health disorders had a global burden, recently estimated at 13%, broadening the magnitude of personal, social, and economic mental disorders costs [17] and increasing health system requirements, which are already burdened and focused in specialties, with the integration of care in the PHC environment as one of the solutions to heath care needs [13]. The CoCM is developed through interprofessional care actions in the PHC, including a behavioral health care manager, who is trained to work together with PHC professionals, and assessment about mental health needs of the specified population with the review of results through supervision of a designated psychiatric consultant [15,18]. Nevertheless, health care actions and services with the offer of care plans focused on evidence related to ordinary mental disorders such as depression and anxiety in the primary care background [19].

### The Present Study

In Brazil, the guiding document for matrix support in mental health proposed training, monitoring, and evaluation of PHC actions and inclusion of mental health actions in the health information system [20]. In this context, the NASF was created in 2008 and renamed in the revised version of the National Primary Care Policy (NPCP) in 2017 to “Expanded Family Health and Primary Care Center” (NASF-AB). However, the NPCP from 2017 extinguished the funding and disconnected NASF from the National Registry of Health Establishments, making NASF invisible to the SUS and changing the organizational matrix of the Brazilian PHC.

The collaborative care of the matrix support was considered in the last two cycles of the external assessment of the Program for Improving Access and Quality of Primary Care (PMAQ-AB) and included NASF teams: 1813 (46.5%) in the second cycle (2013–2015) and 4110 (93.2%) in the third cycle (2015–2019). The adherence of cities to PMAQ-AB was voluntary, and the number of PCTs contracted in the program constantly increased from 17,483 (51.4% of total teams) in the first cycle to 38,865 (96.4%) in the third cycle [21,22].

The alignment between sociotechnical dichotomies of mental health crisis care of PCNs and concepts of risk and vulnerability are challenging for the community, territorial, and intersectoral care network [23,24].

In this context, the crisis, gaps, and weaknesses of mental health services in Brazil are noticeable, specifically regarding the provision of collaborative care The underfunding derived from the structural changes of SUS PHC through PNAB 2017 and the lack of funding through the Prevent Brazil Program, which extinguished the variable Basic Care Minimum Spent, intensified the impairment; therefore, compromising collaborative care processes at the Primary Care point of the Psychosocial Care Network, mainly in small municipalities [25,26]. However, the technical note no. 3/2020 from the Department of Family Health of the Secretary of Primary Care of the Ministry of Health unlinked the composition of multidisciplinary teams from the typologies of NASF-AB teams and revoked normative instruments for parameters and costing, communicating that there will be no accreditation of new teams. Such act can result in dismissals of professionals and extinction of the NASF-AB with an impact on the PHC resolution [27], such as the care for people with mental and behavioral disorders.

Considering that health systems around the world have introduced pay-for-performance (P4P) programs to encourage health professionals to improve quality, use of care, and efficiency [28], in 2011, the Brazilian Ministry of Health developed a proposal to evaluate the quality of PHC services at the national level through the SUS Access and Quality Improvement Program (PMAQ-AB) [28,29]. PMAQ-AB was the first national P4P scheme and one of the largest in the world [22] conducted by the association between collaborative care processes or mental health matrix support induced by the PMAQ-AB in Brazilian municipalities.

PMAQ-AB was implemented in three cycles, the first (November 2011 to March 2013), second (April 2013 to September 2015), and third round (October 2015 to December 2019) [28]. PMAQ-AB results provided information for financial transfers to local health teams as an incentive proportional to performance over time [29]. Collaborative care through the provision of processes by NASF-AB teams had performance evaluations in the last two rounds. In Brazil, through the offer of matrix support from the NASF-AB teams evaluated by the PMAQ-AB, an overview of the offer of “integrated care” and “collaborative care” was possible. These terms are used to describe the integration of mental health in PHC [30].

The theoretical model shown in Figure 1 presents the integration of mental health into PHC through NASF collaborative care actions, induced and evaluated by the PMAQ-AB, with effects on essential attributes of the integrality of PHC. Such actions also favor longitudinality and coordination, especially in small and non-capital cities. Nevertheless, considering the processes for integrating mental health into PHC and throwbacks and threats to the effectiveness of the NPCP, National Mental Health Policy (NMHP), and National Policy on Drugs (NPD), we aimed to analyze the association of criteria, concordance flows, and effectiveness between PCT and NASF for collaborative mental health care in Brazilian capitals and non-capitals, considering the processes offered at the primary care point of the Psychosocial Care Network of SUS in municipalities with different structures and population sizes.

## 2. Methods

This cross-sectional study was based on secondary data and followed the Strengthening the Reporting of Observational Studies in Epidemiology [31].

The study covered the PCT of 5324 Brazilian cities of all 27 federation units (including the Federal District). The original research sample with teams’ voluntary adhesion was stratified for all capital and non-capital cities in all regions of Brazil. Table 1 was based on data from the third cycle (2015–2019) of the external evaluation of PMAQ-AB [32], Atlas of Human Development in Brazil [33], e-Gestor (Information and Management of Primary Care) [34], and social vulnerability index [35]. The sample was composed of 4110 participants, including 3883 professionals from NASF teams. Most participants (64.7%) were pharmacists that responded to Module IV of the third PMAQ-AB (interview with a professional from NASF team and verification of documents at the health unit).

Agreement criteria and flows between PCTs and NASF for mental health care in PHC and specialized services were considered the dependent variables. Independent variables were related to actions of collaborative care of NASF: (1) follow-up of cases with Psychosocial Care Centers (CAPS), (2) support to PCT in the management of psychopharmacotherapy, (3) offer of psychopharmacotherapy with other therapeutic actions, (4) support the approach and actions of PCT in the care of users of psychoactive substances, and (5) offer of activities to prevent the use of psychoactive substances in schools or other areas.

Independent and dependent variables were coded and named in the analysis according to parameters defined in the theoretical model. The SPSS software (version 22.0, IBM Corp. Armonk, NY, USA) was used considering the frequency distribution of teams in capital and non-capital cities of Brazil.

### Data Analysis

Variables were dichotomized (yes and no), and responses to each item were added and divided by the total number of participants. Pearson’s chi-square (χ^2^) test was used to verify the association of agreement criteria and flows between PCT and NASF for mental health care in PHC and specialized services and other NASF collaborative care actions. 

A multiple Poisson regression adjusted by independent variables assessed associations, prevalence ratio, and 95% confidence intervals. We also detected outliers, heterogeneity of variance, collinearity, and dependence on observations [36]. Statistical significance was set at *p* < 0.05.

The research ethics committees of the Alcides Carneiro University Hospital of the Federal University of Campina Grande (No. 5182) and Onofre Lopes University Hospital of the Federal University of Rio Grande do Norte (No. 5292) approved the study protocol. The study was conducted according to Resolution 466/12 of the National Health Council and the Declaration of Helsinki.

## 3. Results

Bivariate and Poisson regression analyses (significance level of 5%) were performed to analyze the association of agreement criteria and flows between PCT and NASF for mental health care in PHC and specialized services, considering the difference between capital and non-capital cities of Brazil. These analyses highlight the long-term effect of two PMAQ-AB cycles (2014–2015 and 2015–2019) that induced NASF actions to integrate mental health into PHC through the provision of mental health collaborative care processes (service organization and professional practice) in the FHS. This reinforces the potential of NMHP, NPD, and WHO guidelines to redirect the care model according to the organization of the PCN.

The agreement criteria and flows between PCT and NASF for mental health care in PHC and specialized services were associated with (1) actions of matrix support and mental health collaborative care via NASF and (2) type of city (capital and non-capital) (Table 2).

The agreement criteria and flows between PCT and NASF with collaborative care actions were 3.84-fold higher in capital cities than in non-capital cities. Capital cities in Brazil have the highest number of services and care points of Health Care Networks. In addition, capital cities present a higher number of PCT and FHS coverage, municipal human development index, and social vulnerability index than non-capital cities. Therefore, this situation has repercussions for NASF actions.

The prevalence of follow-up of cases with PCC that performed the agreement between PCT and NASF was 1.52-fold higher than those who did not perform the follow-up.

Additionally, the prevalence of agreement criteria and flows between PCTs and NASF for mental health care in PHC and specialized services in teams that conducted actions to support PCTs in the management of psychopharmacotherapy and offer of psychopharmacotherapy with other therapeutic actions was 1.73-fold and 1.42-fold higher, respectively, than in teams not performing these actions.

Furthermore, the prevalence of agreement criteria and flows between PCTs and NASF for care in primary and specialized care was 1.49-fold higher in PCTs who received actions to support the approach and care of users of psychoactive substances than those who did not receive this action. 

Actions to offer activities to prevent the use of psychoactive substances in schools or other areas showed a positive association 1.63-fold higher to perform the agreement criteria and flows between PCTs and NASF for mental health care in PHC and specialized services than teams not performing this activity.

The agreement criteria and flows between PCTs and NASF teams favor mental health care in PHC and specialized services demonstrated negative associations with a protective factor (PR < 1) in the following exposures: (1) 97% [95% to 98%] for capital cities, (2) 94% [92% to 96%] for actions of follow-up of cases with PCC, (3) 93% [91% to 96%] for actions to support PCT in the management of psychopharmacotherapy, (4) 98% [95% to 100%] to offer psychopharmacotherapy with other therapeutic actions, (5) 89% [85% to 92%] for actions to support PCT in the care process for users of psychoactive substances, and (6) 93% [90% to 95%] for actions to offer activities to prevent the use of psychoactive substances in schools or other areas (Table 3).

Exposure variables were the actions of NASF-AB and PCTs in territorial-based services and PHC of the PCN of the SUS in Brazilian cities. The negative associations observed highlight the importance and need of PCC teams via matrix support and collaborative care to integrate mental health into PHC. This support is also needed to manage clinical protocols and therapeutic guidelines via intersectionality of actions, both for psychopharmacotherapy and other non-pharmacological psychosocial care actions for alcohol and other drug users.

## 4. Discussion

The regression analysis adjusted for the association of agreement criteria and flows between PCT and NASF for collaborative mental health care in the primary and specialized care of the PCN showed negative associations. A protective factor was observed for actions of NASF teams in capital cities, follow-up of cases with PCC, support to PCT in the management of psychopharmacotherapy, offer of psychopharmacotherapy with other therapeutic actions, support to PCT in the care process for users of psychoactive substances, and offer of activities to prevent the use of psychoactive substances in schools or other areas. These associations may help diagnose the situation of mental health integration into PHC via collaborative care of the NASF induced by the long-term effects of PMAQ-AB in Brazil.

The dichotomous variables of the PMAQ-AB database can be considered a study limitation. On the one hand, the database highlights matrix support actions and collaborative care via NASF. On the other hand, it does not identify specific variables related to sociodemographic aspects, age group, health promotion for all classifications of mental disorders, and equity of access for people with mental and behavioral disorders. Additionally, data regarding mental health collaborative care actions were self-reported only by NASF professionals that joined the third cycle of the external evaluation of PMAQ-AB in capital and non-capital cities of Brazil.

The analysis of results from this approach can be supported by other studies that showed low percentages of implementation of mental health care in PHC (at the national level); persistence of regional inequalities of access in RAPS (mainly in high-complexity hospitals in the national context); gaps in health promotion strategies, care management, and pharmaceutical assistance; and the offer of processes in Primary Care in some RAPS settings with potential for better performance and longitudinal follow-up [37,38]. Gaps in collaborative mental health care can be filled through a positive association. For instance, there is a protective factor between the implementation of matrix support and the improvement in the detection and treatment of mental disorders in primary care, suggesting effects of this primary mental health care are favored by intensifying local organizational support for clinical integration among RAPS workers, including those at the Primary Care point [39].

The demand for mental health services in Brazil increased despite incentives for permanent education in health and guidelines from international and national health agencies, including state and city health departments with flow criteria for treatment. Additionally, NASF professionals recognized the lack of instruments and strategies to quantify and organize this demand [40]. Nevertheless, PCTs receiving support from NASF reported associations between social problems (e.g., low income, unemployment, and large number of inhabitants per household) and complaints related to mental health [40].

The Brazilian findings support the international evidence about the cooperative model based on the PHC measure, which uses proactive data collection to offer plans related to care focused on the patient, including clinical and systematic measures of longitudinal care, which are guided to scanning, assessment, and answers to individualized results, such as the seriousness of symptoms and the extent of goals in the long term, and through practical guidelines. These actions foster the enhancement of access and quality of integrated care of mental health in the PHC with the other levels of assistance, fostered by P4P to health professionals [13]. Towards this direction, there are initiatives, such as Improving Access to Psychological Therapies (IAPT), which belongs to National Health Service of the United Kingdom, the Dutch initiative to Depression, and the Australian model TrueBlue, although they do not reach all patients with mental disorders and do not include, on a daily basis, most health aspects [13]. Although positive effects are present in the PHC model, such as users’ satisfaction and effectiveness of care, there are conflicts among professionals related to treatment time length and full-flow design to make collaborative work [18].

The results in Brazil, a middle-income country, but with a wide range of inequalities and iniquities, are aligned to evidence a shortage of health professionals when compared to high-income countries; although the collaborative care is based in evidence to the treatment of ordinary mental health problems in the PHC, it is advisable that obstacles be overcome with appropriate approaches and operationalized in environments with scarce resources [19].

The effects of PMAQ-AB highlight the diagnosis of mental health integration into PHC and reveal aspects related to access and quality of collaborative care processes via NASF. The protective factor in the analysis of agreement criteria and flows between PCTs and NASF for mental health care in capital cities can also be related to a better specialized structure for mental health and involvement of professionals from different areas in PHC, which is uncommon in small and medium-sized cities [40].

Since the first cycle (2011), the effects of PMAQ-AB on the evaluation and monitoring of PCTs induced qualification, planning, schedule of structures and processes, and results prioritizing social and programmatic vulnerabilities in the NPCP. Financial incentives for PHC were also observed, such as the implementation of PCTs specific for a territory or population (riverside, river, and prison care), expansion of NASF, implementation of Street Offices, promotion and prevention actions (Health at School Program), and home care (Better at Home Program) [22]. The PMAQ-AB also participated in other initiatives of the Ministry of Health, such as the Basic Health Unit Requalification Program, investment and implementation of a PHC information system (e-SUS), telehealth (Telehealth Brazil Network), Appreciation of Professionals of Primary Care Program, provision of professionals via the More Doctors Program, and the increase of medical and multi-professional residency vacancies in PHC [22]. These initiatives complement each other for systematization and coordination of the PHC and improvement of the infrastructure of basic health units (i.e., implementation of electronic health records in e-SUS primary care since 2016).

Nevertheless, based on the second cycle (2013–2015) of PMAQ-AB, the incentive to P4P did not ensure the adherence of vulnerable populations from priority regions to resources and investment in teams [41]. Furthermore, despite satisfactory results for capital cities, risk factors for problem solving and quality of care offered in PHC were observed in the second cycle, such as a deficit in training and qualification of professionals for better integration between PCTs and NASF (especially collaborative care), monitoring of the therapeutic project, planning based on health indicators, and heterogeneity of actions between the Ministry of Health and state and municipal health departments [41]. Seen in these terms, it was shown that the growth of payment models based on values that compensate the health systems and professionals to achieve results and not based on service volumes is imperative to enhance the quality of mental health care [13].

The results of this study demonstrate a protective factor for collaborative care actions offered by PCC in capital cities, which corroborate investments of the NMHP and NPD. PCC are strategic devices of specialized reference that perform territorial, community, and collaborative care actions [24]. Before NASF, programmatic guidelines guided the PCC to offer these actions to the PCTs, while matrix support teams were created when PCC was insufficient [20].

In addition, the configuration of collaborative care in the PCN through actions of NASF allowed the understanding of the performance of processes of PCTs and NASF (organization of services and clinical practice) in risk areas with territorial problems. The study used the perspective of planning, programming, and supply strategies, given the systematization of clinical guidelines and therapeutic protocols in the PCN of SUS and considering the deinstitutionalization of the Brazilian psychiatric reform. Therefore, issues regarding processes, structure, epidemiology, and policy are essential for the operationalization of collaborative care, in addition to invisible resources (e.g., interpersonal and interprofessional relationships) in which principles of decentralization, integrality, and equity of SUS are materialized and reinvented in practice [42]. Data also revealed the potential of policies, programs, PCN, and mental health actions for changing traditional patterns and the effects of PMAQ-AB in the collaborative and interprofessional care process for integrating mental health into PHC in Brazil. 

The use of an evidence base, such as the Mental Health Gap Action Program of WHO, is a protective factor for managing psychopharmacotherapy in the PHC. This program encourages mental health services in the PHC of low- and middle-income countries [43]. Despite guidelines, the translation, dissemination, and implementation of evidence-based mental health policies are slow in low-income countries [44].

The collaborative care management or matrix support for psychopharmacotherapy in Brazil is oriented by guidelines (Primary Care Notebooks and pharmaceutical practices in the NASF-AB), which includes guidance for the rational use of medicines and support for people with psychological distress or mental disorders [28]. In this context, most PCT professionals perceive the abuse of psychotropic drugs and claim cultural heritage for the ignorance of users and side effects [40].

Another protective factor for managing psychopharmacotherapy users in the FHS is the mitigation of social medicalization subsidized by the biomedical model that subordinates the PCT work process. This reveals the process of medicalization in the biologicist PHC model that transfers responsibility to individual factors and is centered on clinical practice, not responsible for sociopolitical issues of the territory [45].

In the field of care actions and considering PHC complexity, the results of our study indicate the use of soft technologies (i.e., offer of other non-pharmacological therapeutic actions for mental and behavioral disorders) as a protective factor for capital cities. These actions were not listed or explained in the questionnaire of the third cycle, despite the inclusion of actions in clinical guidelines and evidence-based therapeutic protocols (i.e., home visits, reception, shared consultation, interprofessional and intersectoral actions, supported self-care, clinical case discussion, risk stratification, operating group, Singular Therapeutic Project, care management, and case management). Despite this protective factor, the potential of soft technologies for people in psychological distress is still underused. Moreover, the low professional qualification limits the diagnosis, response to psychosocial treatment, and rehabilitation [46].

Offering support to PCTs in the care process of users of psychoactive substances was also a protective factor for capital cities. Therefore, we may infer the need for better conditions of structure and processes, including a psychosocial care center specialized in alcohol and other drugs (PCC for Alcohol and Drugs (PCCAD)) and other structuring equipment of PCNs (i.e., Street Clinic, Shelter Unit, and Community Centers), whose implementation depends on the epidemiological profile and population size.

Despite the protective effect for the third cycle of PMAQ-AB, throwbacks of the Federal Government must be highlighted, such as changes in the NPD, exclusion of harm reduction care for people with substance abuse, and election of abstinence as the only possible treatment. The replacement of absentee models strengthened biomedical psychiatrization through involuntary hospitalizations, judicialized care through coercive forces, and moralized treatments by prioritizing private institutions of religious and spiritual orientation (i.e., Therapeutic Communities) [47].

The risk factors of these throwbacks can be related to the ineffectiveness of clinical practices and absenteeism of PHC users and specialized services. A crisis in the penitentiary system is also possible because of overcrowding and the creation of new services for psychosocial rehabilitation due to misunderstandings regarding the war on drugs [47].

Considering the need for interconnection of collaborative care actions in the agreement criteria and flows between PCTs and NASF, we highlight the linkage between follow-up actions and PCCAD, in which professionals of services identified an underutilization of the PHC approach for reducing harms and PCT and matrix support professionals indicated barriers in the referral to some PCCAD [47].

The module IV of PMAQ-AB elected the offer of promotion and prevention activities for the use of psychoactive substances instead of care processes for other mental disorders. This position neglected the burden of other mental disorders, age groups in the PHC, deinstitutionalization, and psychosocial rehabilitation strategies, therefore, justifying the protective factor for capital cities.

Proposals for integrating PCN devices into the PHC may facilitate the identification of prevention and promotion of mental health for better articulation of actions with access, treatment adherence, and rehabilitation according to individual and collective needs [40]. Mental health promotion and prevention actions are included in NASF guidelines [7] due to the burden of mental disorders and insufficient provision of care to close the mental health gap in low- and middle-income countries. Nevertheless, these actions are still incipient in most health systems [44]. Although countries reported increased mental health promotion and prevention programs (52% in 2020), 31% did not have assured human and financial resources, 27% did not plan, and 39% did not document the progress and impacts [10].

These Brazilian findings support the produced evidence about the context in the United States of America, where the necessity of implementation of PHC at a large scale is highlighted by the fact that the system of specialized service in mental health is not able to meet health service demands and, even before the COVID-19 pandemic, the access to mental health care was already limited because of the concentration of mental health specialists in individualized practices in urban areas, which lead to geographical barriers, encumbrance of treatment costs, and isolation for the effectiveness of mental health teams [15]. In this context, the CoCM offers specialized knowledge for the PHC scenario; and even if the number of mental health appointments for PHC rise, its integrated and coordinated nature of the given care offer reduces the inevitable use and the PHC and system demands in general [15].

The effects of the PMAQ-AB corroborate analyses of barriers and threats to mental health, including neglection of mental health promotion and protection in community services for mental health [48]. Throwbacks and crises of the NMHP compromise guidelines of the Brazilian psychiatric reform and promote a new edition that encourages psychiatric hospitalization, decouples NMHP from NPD, emphasizes the funding of therapeutic communities, and prohibits issues related to alcohol and other drugs in the psychiatric reform [49].

Nevertheless, the funding and management plan included in the Previne Brazil Program practically annihilated the NASF-AB. The Federal Government allowed state and city managers to adopt any multiprofessional care model. Consequently, the city and state Health Secretariats must define the multidisciplinary teams and the minimum workload.

## 5. Conclusions

This study evidenced the effectiveness of the criteria and agreement flows for collaborative mental health care, based on the theoretical model proposed for the NASF matrix support process. Such model impacts on planning, programming, provision of promotion, prevention processes, and on mental health care, through the diagnosis of the health situation and pharmacovigilance in the RAPS organization of SUS in Brazil. The long-term effects of PMAQ-AB were positive and improved collaborative mental health care through interprofessional actions at the PHC point of RAPS, effecting the NMHP and NPD according to WHO guidelines, de-institutionalization, and rehabilitation processes of psychosocial development of the Brazilian psychiatric reform. In addition, the protective factor of macrosocial determinants for the structure of capitals in relation to medium and small cities in Brazil was highlighted. The Brazilian Federal Government has a neoliberal agenda, whose conflicts of interest allow attacks and threats to guarantees of national health policies. This affects the NMHP and NPD via underfunding strategies and weakening and dismantling NASF and PCN, compromising the provision of mental health collaborative care processes in the PHC.

Last, investments for integrating mental health into PHC, providing professional training and Permanent Education in Health focused on social determinants of health, and promoting mental health according to clinical guidelines and protocols of health agencies are challenging, aiming to mitigate inequities, inequalities of access, care gaps, differences in the structure and supply of the mental health care processes between small, medium, and large municipalities, in a country with regional disparities and territorial dimension, such as Brazil.

Although there is evidence about CoCM mental health in Brazil, the strong points of this study are presented through detailed description on the criteria of agreement and flows of CoCM in the Brazilian PHC, in a P4P scheme, through the data analysis of programs and service processes offered nationwide, in a country with continental dimensions such as Brazil. However, there are theoretical contributions to rebuild policies, programs, guidelines, and mental health care technologies and training of skills and abilities in the education of health professionals; and there are practical implications for the planning and management of mental health processes through the integration of care based on the person in different PHC contexts.

Considering study limitations, randomized clinical essays should be performed to confirm evidence of the CoCM relation in the PHC, with the classification of mental and behavior disorders, health social determinants, care technologies, and the promotion of mental health.

## Figures and Tables

**Figure 1 ijerph-19-15148-f001:**
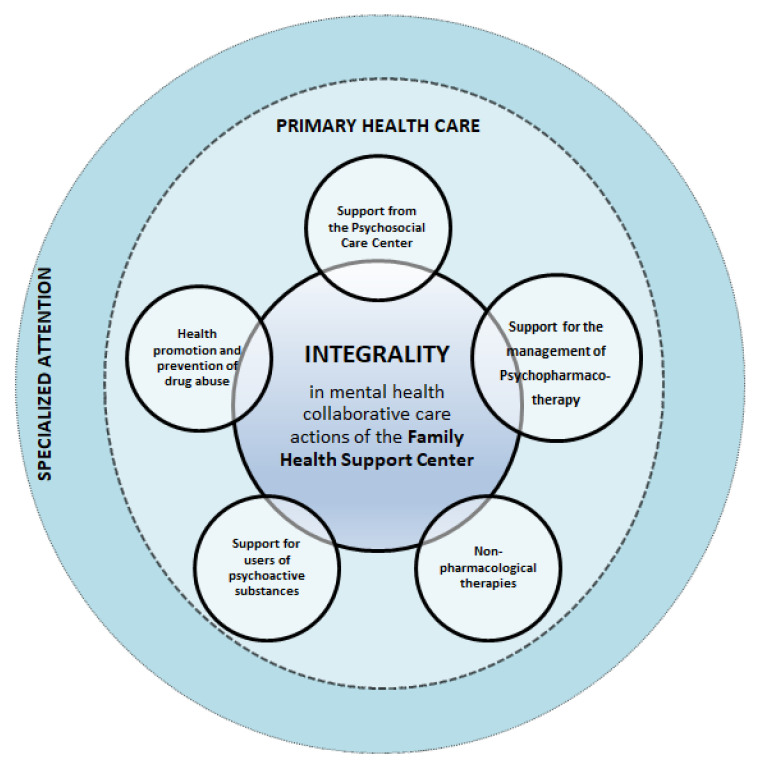
Theoretical model of mental health collaborative care of the Family Health Support Center, induced by the National Program for Improving Access and Quality of Primary Care in Brazil.

**Table 1 ijerph-19-15148-t001:** Study scenario.

Region/ Federation Units	Population (Inhabitants)	Demographic Density (Inhab/km^2^)	Percentage of Population in Rural Areas (%)	Number of Primary Care Teams	FHS Coverage (%)	Life Expectancy at Birth (Years)	Municipal Human Development Index	Social Vulnerability Index (%)
Midwest								
Federal Districty	3,036,006	525.26	3.42	321	36.43	78.37	0.850	0.250
Goiás	6,779,027	19.93	9.71	1433	65.68	74.34	0.799	0.232
Mato Grosso	3,297,200	3.65	18.20	727	69.34	74.47	0.774	0.223
Mato Grosso do Sul	2,647,973	7.52	14.36	598	71.93	75.80	0.766	0.190
Northeast								
Alagoas	3,369,183	120.98	26.36	873	75.92	71.97	0.683	0.327
Bahia	15,324,591	27.14	27.14	3674	73.37	73.71	0.714	0.288
Ceará	9,021,470	60.59	24.91	2545	83.12	74.07	0.735	0.259
Maranhão	6,964,705	20.98	36.92	2131	84.50	70.85	0.687	0.347
Paraíba	4,002,758	70.88	24.63	1418	94.66	73.53	0.722	0.300
Pernambuco	9,415,052	96.00	19.83	2345	77.52	74.27	0.727	0.320
Piauí	3,219,953	12.80	34.23	1343	99.82	71.23	0.697	0.271
Rio Grande do Norte	3,507,564	66.42	22.19	1038	79.05	75.96	0.731	0.271
Sergipe	2,288,163	104.39	26.48	639	84.99	72.92	0.702	0.303
North								
Acre	816,603	4.98	27.44	224	78.63	74.25	0.719	0.347
Amapá	791,788	5.54	10.23	158	62.16	74.19	0.740	0.239
Amazonas	3,918,175	2.51	20.91	696	54.81	72.14	0.733	0.328
Pará	8,328,271	6.70	31.52	1.521	59.10	72.29	0.698	0.282
Rondônia	1,796,762	7.56	26.45	377	68.93	71.53	0.725	0.197
Roraima	466,021	2.08	23.45	129	73.20	71.84	0.752	0.253
Tocantins	1,537,879	5.54	21.20	508	94.68	73.65	0.743	0.247
Southeast								
Espirito Santo	4,012,291	87.06	16.60	725	59.05	76.02	0.772	0.214
Minas Gerais	21,110,383	35.99	14.71	5527	80.21	77.49	0.787	0.202
Rio de Janeiro	16,723,083	381.97	3.29	2913	58.66	76.48	0.796	0.277
São Paulo	45,103,052	181.71	4.06	5327	39.73	76.26	0.826	0.237
South								
Paraná	11,310,996	56.75	14.67	2333	65.56	75.55	0.792	0.184
Rio Grande do Sul	11,310,085	42.10	14.90	2113	59.93	75.95	0.787	0.210
Santa Catarina	6,988,533	73.13	16.01	1791	79.57	76.97	0.808	0.127

FHS: Family Health Strategy.

**Table 2 ijerph-19-15148-t002:** Bivariate associations of agreement criteria and flows between primary care teams (PCT) and family health support center (NASF) for mental health care in PHC and specialized services, considering the difference for capital and non-capital cities of Brazil (2018).

Variable		Criteria Consesus and Flows between PCT and NASF		Total	*p*-Value	PR [95% CI]
		Yes	No			
City type	Capital	97.51% (n=313)	2.49% (n = 8)	100.00% (321)	<0.001	3.84 [1.92; 7.68]
	Non-capital	90.48% (n = 3223)	9.52% (n = 339)	100.00% (n = 3562)		
Follow-up of cases with PCC	Yes	93.94% (n = 2682)	6.06% (n = 173)	100.00% (n = 2855)	<0.001	1.52 [1.37; 1.69]
	No	83.07% (n = 854)	16.93% (n = 174)	100.00% (n = 1028)		
Support to PCT in psychopharmacotherapy	Yes	94.63% (n = 2852)	5.37% (n = 162)	100.00% (n = 3014)	<0.001	1.73 [1.54; 1.94]
	No	78.71% (n = 684)	21.29% (n = 185)	100.00% (n = 869)		
Offer of psychopharmacotherapy with other therapeutic actions	Yes	93.53% (n = 2963)	6.47% (n = 205)	100.00% (n = 3168)	<0.001	1.42 [1.30; 1.55]
	No	80.14% (n = 573)	19.86% (n = 142)	100.00% (n = 715)		
Support the approach and actions of the PCT in the care of users of psychoactive substances	Yes	93.82% (n = 3236)	6.18% (n = 213)	100.00% (n = 3449)	<0.001	1.49 [1.37; 1.62]
	No	69.12% (n = 300)	30.88% (n = 134)	100.00% (n = 434)		
Offer of activities to prevent the use of psychoactive substances in schools or other areas	Yes	94.31% (n = 2915)	5.69% (n = 176)	100.00% (n = 3091)	<0.001	1.63 [1.46; 1.80]
	No	78.41% (n = 621)	21.59% (n = 171)	100.00% (n = 792)		

PCC (psychosocial care centers); PCT (primary care teams); PR (prevalence ratio); CI (confidence interval).

**Table 3 ijerph-19-15148-t003:** Poisson regression analysis of consensus criteria and flows between primary care teams (PCT) and family health support center (NASF) for mental health care in PHC and specialized services in capital cities of Brazil (2018).

Parameter of Affirmative Answers	β	Standard Error	Wald 95% CI		Hypothesis Test			Exp (β)	Wald 95% CI for Exp (β)	
			Lower	Upper	Wald Chi-Squared	df	Sig.		Lower	Upper
Intercept	0.38	0.02	0.34	0.41	378.26	1	<0.001	1.46	1.40	1.51
Type of city (capital)	−0.03	0.01	−0.05	−0.02	13.20	1	<0.001	0.97	0.95	0.98
Follow-up of cases with PCC	−0.07	0.01	−0.09	−0.05	40.12	1	<0.001	0.94	0.92	0.96
Support to PCT in psychopharmacotherapy	−0.07	0.01	−0.10	−0.04	28.39	1	<0.001	0.93	0.91	0.96
Offer of psychopharmacotherapy with other therapeutic actions	−0.02	0.01	−0.05	0.00	2.69	1	0.101	0.98	0.95	1.00
Support the approach and actions of the PCT in the care of users of psychoactive substances	−0.12	0.02	−0.16	−0.08	38.77	1	<0.001	0.89	0.85	0.92
Offer of activities to prevent the use of psychoactive substances in schools or other areas	−0.08	0.01	−0.10	−0.05	35.96	1	<0.001	0.93	0.90	0.95

PCC (psychosocial care centers); PCT (primary care teams); CI (confidence interval); df: degrees of freedom.

## Data Availability

Not applicable.

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
