# Peer review of "Effectiveness of Agreement Criteria and Flows of Collaborative Care in Primary Mental Health Care in Brazil"

_ijerph, 2022, doi:10.3390/ijerph192215148_

Round 1

Reviewer 1 Report (Previous Reviewer 3)

The authors were very responsive with regard all my observations. I believe that now the manuscript is ready to be accepted.

Author Response

Dear Reviewer, thank you for spending your time reviewing our manuscript. We deeply appreciate your indication to publish our manuscript. Attached you can find the final version of our manuscript. 

Best reagards, 

Valdecir Carneiro.

Reviewer 2 Report (New Reviewer)

The article refers to a significant issue of mental health collaborative care. The authors used the secondary data but presented the analyses reliably and came to interesting conclusions. From the methodological and analytical point of view, the article does not raise any objections. However, I would suggest strengthening its scientific side by:

1.       Presenting general scientific literature relating to mental health collaborative care. The authors thoroughly analyse the literature relating to the situation in Brazil. However, the background should consider international scientific achievements, not only WHO reports and Brazilian literature. Such a part related to theoretical background could be placed before the "The present study" section.

2.       In developing my suggestion 1, it would be valuable to refer the obtained results to the general scientific literature relating to mental health collaborative care in the Discussion section. Currently, the Discussion is instead a summary of the results and the authors' deliberations on this subject without a scientific basis.

3.       After discussing the results with the general scientific literature relating to mental health collaborative care, it will be possible to present the added value of this work by identifying theoretical and practical implications.

4.       It would also be worth indicating future directions of research.

Author Response

We appreciate all valuable comments on the manuscript. We have carefully considered all points raised by the reviewer and modified our manuscript accordingly to address your comments. Suggestions significantly improved our manuscript. Please, do not hesitate to contact us if you have any questions.

Attached you can find the last version of our manuscript and below you can find our responses to your comments. We emphasize that all alterations made from the last version are highlighted in red in the body of the manuscript text.

RESPONSE TO SECOND REVIEWER'S COMMENTS (02):

  1. Presenting general scientific literature relating to mental health collaborative care. The authors thoroughly analyse the literature relating to the situation in Brazil. However, the background should consider international scientific achievements, not only WHO reports and Brazilian literature. Such a part related to theoretical background could be placed before the "The present study" section.

Answer: Suggestion accepted with the input of text and references (before the section, “The present study”, page 2)

The Collaborative Care Model (CoCM) is well known strategy, based on evidence, grounded on a Chronical Care Model (CCM and originally made to describe complex evidence related to the care level to i) accomplish multiprofessional approach; ii) improve interprofessional communication; iii) enhance programmed health follow-ups; and iv) make the structured  management plan [1-3]. Regarding health care procedures, those apply care management approach by systematic monitoring and links between primary care and mental health professionals, offering patients: 1) mental health exams; 2) interventions; 3) care facilitations; and 4) follow-ups [4].

In the current epidemiological transition, impacts prevail, before and after the COVID-19 pandemic, mental and behavior health disorders global burden, recently estimated in 13%, broadening the magnitude of personal, social and economic mental disorders costs [5] and increasing health system requirements, which are already burdened and focused in specialties, with the integration of care in the PHC environment as one of the solutions to heath care needs [1]. The CoCM is developed through interprofessional care actions in the PHC, including a  behavioral health care manager, who is trained to work together with PHC professionals; and assessment about mental health needs of the specified population with the review of results through supervision of a designated psychiatric consultant [3,6]. Nevertheless, health care  actions and services with the offer of care plans focused on evidence related to ordinary mental disorders , like depression and anxiety in the primary care background [7].    

  1. In developing my suggestion 1, it would be valuable to refer the obtained results to the general scientific literature relating to mental health collaborative care in the Discussion section. Currently, the Discussion is instead a summary of the results and the authors' deliberations on this subject without a scientific basis.

Answer: Suggestion accepted with the input of text and references (before the fourth and seventh paragraph on the page 09; and fifteenth paragraph on page 12 of the ‘Discussion’ section)

The Brazilian findings support the international evidence about the cooperative model based on the PHC measure, which uses proactive data collection to offer plans related to care focused on the patient, including clinical and systematic measures of longitudinal care, which are guided to scanning, assessment, and answers to individualized results, such as the seriousness of symptoms and the extent of goals in the long term, and through practical guidelines. These actions foster the enhance of access and quality of integrated care of mental health in the PHC with the other levels of assistance, fostered by P4P to health professionals [1]. Towards this direction, there are initiatives, such as Improving Access to Psychological Therapies (IAPT), which belongs to National Health Service of the United Kingdom, the Dutch initiative to Depression and the Australian model TrueBlue, although they do not reach all patients with mental disorders and do not include, on a daily basis, most health [1]. Although positive effects are present in the PHC model, such as users’ satisfaction and effectiveness of care, there are conflicts among professionals related to treatment time length and full-flow design to make collaborative work [6].

The results in Brazil, a middle-income country, but with a wide range of inequalities and iniquities, are aligned to evidence of a shortage of health professional when compared to high-income countries; although the collaborative care is based in evidence to the treatment of ordinary mental health problems in the PHC, it is advisable that obstacles be overcome with appropriate approaches and operationalized in environment with scarce resources [7].

Seen in these terms, it was shown that the growth of payment models based on values that compensate the health systems and professionals to achieve results  and not based on service volumes is imperative to the enhance of quality of mental health care [1].

These Brazilian findings support the produced evidence about the context in the United States of America, where the necessity of implementation of PHC in large scale is highlighted by the fact that the system of specialized service in mental health is not able to meet health service demands and, even before the COVID-19 pandemic, the access to mental health care was already limited because of the concentration of mental health specialists in individualized practices in urban areas, which lead to geographical barriers, encumbrance of treatment costs and isolation for the effectiveness of mental health teams [3]. In this context, the CoCM offers specialized knowledge for the PHC scenario; and even if the number of mental health appointments for PHC rise, its integrated and coordinated nature of given care offer reduces the inevitable use and the PHC and system demands in general [3].

  1. After discussing the results with the general scientific literature relating to mental health collaborative care, it will be possible to present the added value of this work by identifying theoretical and practical implications.

Answer: Suggestion accepted with the input of text (at the end of the ‘Conclusion’ section, page 12-13)

Although there is evidence about CoCM mental health in Brazil, the strong points of this study are presented through detailed description on the criteria of agreement and flows of CoCM in the Brazilian PHC, in a P4P scheme, through the analysis of national data basis, which comes from the assessment of offer of program and service processes, in a country with continental dimensions, like Brazil. However, there are theoretical contributions to rebuild policies, programs, guidelines and mental health care technologies and training of skills and abilities in the education of health professionals; and practical implications for the planning and management of mental health processes through the integration of care based on the person in different PHC contexts.

  1. It would also be worth indicating future directions of research.

Answer: Suggestion accepted with the input of text (at the end of the ‘Conclusion’ section, page 13)

Considering study limitations, randomized clinical essays should be performed to confirm evidence of the CoCM relation in the PHC, with the classification of mental and behavior disorders, health social determinants, care technologies, and the promotion of mental health.

REFERENCES INSERTED ON THE MANUSCRIPT

  1. Kilbourne, A.M.; Beck, K.; Spaeth-Rublee, B.; Ramanuj, P.; O'Brien, R.W.; Tomoyasu, N.; Pincus, H.A. Measuring and improving the quality of mental health care: a global perspective. World Psychiatry 2018, 17, doi: https://doi.org/10.1002/wps.20482.
  2. Curth, N.K.; Brinck-Claussen, U.Ø.; Hjorthøj, C.; Davidsen, A.S.; Mikkelsen, J.H.; Lau, M.E.; Lundsteen, M.; Csillag, C.; Christensen, K.S.; Jakobsen, M.; et al. Collaborative care for depression and anxiety disorders: results and lessons learned from the Danish cluster-randomized Collabri trials. BMC Family Practice 2020, 21, 234, doi:10.1186/s12875-020-01299-3.
  3. Carlo, A.D.; Barnett, B.S.; Unützer, J. Harnessing Collaborative Care to Meet Mental Health Demands in the Era of COVID-19. JAMA Psychiatry 2021, doi:10.1001/jamapsychiatry.2020.3216.
  4. Fiscella, K.A.-O.; McDaniel, S.A.-O. The complexity, diversity, and science of primary care teams. Am Psychol. 2018, 73, doi:10.1037/amp0000244.
  5. Coelho, V.A.A.; Andrade, C.L.T.D.; Guimarães, D.A.; Pereira, P.H.B.; Modena, C.M.; Guimarães, E.A.A.; Gama, C.A.P.D. Ciênc. saúde coletiva. Ciênc. saúde coletiva 2022, 27, doi: https://doi.org/10.1590/1413-81232022275.11212021.
  6. Holmes, A.; Chang, Y.-P. Effect of mental health collaborative care models on primary care provider outcomes: an integrative review. Family Practice 2022, 39, 964-970, doi:10.1093/fampra/cmac026 %J Family Practice.
  7. Unützer, J.; Carlo, A.D.; Collins, P.Y. Leveraging collaborative care to improve access to mental health care on a global scale. World Psychiatry 2020, 19, doi:10.1002/wps.20696.

This manuscript is a resubmission of an earlier submission. The following is a list of the peer review reports and author responses from that submission.

Round 1

Reviewer 1 Report

This study has failed to raise an appropriate research question. The information is messy, and the approaches are too superficial. I do not think this paper meets the standard of IJERPH for publication. 

Reviewer 2 Report

Dear Editor and kind authors,

I think this topic may be of interest.

Nevertheless, I found the reading of this article a bit tiring.

I think the structure of services needs to be better explained.

For example, I do not quite understand whether the primary mental health care services are public or private.

I think the text needs to be simplified a bit and avoid using too many acronyms.

For statistical analysis, I am afraid that making all variables dichotomous simplifies the view of the situation, and it is not clear why Poisson regression was used.

I would also add “drug addiction” among the key words.

I think the following list should be followed to correct the structure of the article:

---Introduction

begun with a statement of the main issues being addressed?

made a clear case as to why the study was needed?

described any necessary background information about the setting for the study?

justified the choice of measures and the sampling strategy?

clearly stated the research questions/hypotheses? This can be done with a bulleted or numbered list.

---Methods

clearly stated the total number of participants and the numbers in each group if there are groups?

clearly stated the response rate?

described the sample in terms of age, sex breakdown, social class or other relevant properties?

clearly stated how the sample was obtained in such a way as to be able to judge its representativeness?

clearly listed the measures used, indicating where possible indices of validity and reliability or giving a citation where these can be found?

described the intervention(s) if appropriate and any control conditions if present?

explicitly described any complex or unusual statistical methods?

---Results

described the results in terms of the answers to the questions/aims presented at the end of the introduction in the same order as they were presented then?

---Discussion

begun the discussion with a summary of the main findings?

related the findings to previous research in terms of whether they support or fail to support the conclusions of that research?

explained how the findings reflect on theory, practice or policy formulation?

examined the limitations of the study, addressing issues such as sample size, sample representativeness, measurement error, measurement bias, whether any intervention was successfully implemented, whether there was contamination between different intervention conditions and ability to generalize from the findings?

attempted to explain apparently anomalous findings?

finished with a paragraph summarizing the main conclusions?

Reviewer 3 Report

The manuscript has as aims to analyze the association of agreement criteria and flows between primary care teams and the Family Health Support Center for mental health collaborative care, considering the difference between capital and non-capital cities in Brazil. It is an interesting paper and I congratulate the authors for the great work done. However, I have some minors concerns on the manuscript:

-        In the introduction section, it is missing that the authors justify the need for the study they propose. What gaps currently exist? In what ways does the supply of mental health processes in primary care have gaps? The authors should be clearer on that aspect

-        I recommend authors include a section “the present study”. This section should be in another paragraph. Please, make sure all objectives are stated in this section and explain the results expected.

-        Methods section needs a reorganization. I recommend authors include a Data Analysis section where authors explain in details data analyses carried out in the present study.

-        Discussion section should be start with the mains of the study in a first paragraph.

-        Please describe the contributions of the study more clarify and extensity.

-        At the end of the manuscript, the practical applications/implications of the study should be explained in details.